# β-Hydroxybutyrate Reduces Body Weight by Modulating Fatty Acid Oxidation and Beiging in the Subcutaneous Adipose Tissue of DIO Mice

**DOI:** 10.3390/ijms26115064

**Published:** 2025-05-24

**Authors:** Violeta Heras, Virginia Mela, Pallavi Kompella, Elena Rojano, Guillermo Paz-López, Lucia Hurtado-García, Almudena Ortega-Gomez, Maria José García-López, María Luisa García-Martín, Juan A. G. Ranea, Francisco J. Tinahones, Isabel Moreno-Indias

**Affiliations:** 1Instituto de Investigación Biomédica de Málaga y Plataforma en Nanomedicina, IBIMA Plataforma BIONAND, 29590 Málaga, Spain; elenarojano@outlook.com (E.R.); guillepl@uma.es (G.P.-L.); lhurtado@uma.es (L.H.-G.); almudena.ortega@ibima.eu (A.O.-G.); mari.jose.baza@gmail.com (M.J.G.-L.); mlgarcia@ibima.eu (M.L.G.-M.); ranea@uma.es (J.A.G.R.); fjtinahones@uma.es (F.J.T.); isabel.moreno@ibima.eu (I.M.-I.); 2Department of Endocrinology and Nutrition, Virgen de la Victoria University Hospital, 29010 Málaga, Spain; 3Center for Biomedical Network Research (CIBER) in Physiopathology of Obesity and Nutrition (CIBEROBN), Carlos III Health Institute, 28029 Madrid, Spain; 4Nucleus, Sarafan ChEM-H, Stanford University, Stanford, CA 94305, USA; kvpallavi@gmail.com; 5Department of Molecular Biology and Biochemistry, Faculty of Sciences, University of Málaga, Bulevar Louis Pasteur, 31, 29010 Málaga, Spain; 6Center for Biomedical Network Research on Rare Diseases (CIBERER), Avenue. Monforte de Lemos, 3-5, Pabellón 11, Planta 0, 28029 Madrid, Spain; 7Department of Medicine and Dermatology, Faculty of Medicine, University of Malaga, 29010 Malaga, Spain; 8Biomedical Magnetic Resonance Laboratory-BMRL, Andalusian Public Foundation Progress and Health-FPS, 41092 Seville, Spain; 9Spanish National Bioinformatics Institute (INB/ELIXIR-ES), Instituto de Salud Carlos III (ISCIII), 28020 Madrid, Spain

**Keywords:** obesity, β-hydroxybutyrate, adipose tissue, SAT, thermogenesis, fatty acid oxidation, weight loss

## Abstract

β-hydroxybutyrate (BHB) serves as an alternative cellular fuel during states of low glucose availability, such as fasting or carbohydrate restriction, when the body shifts to using fats and ketone bodies for energy. While BHB has shown potential metabolic benefits, its mechanisms of action in the context of obesity are not fully understood. In this study, we examined the effects of BHB supplementation on subcutaneous adipose tissue (SAT) metabolism in a diet-induced obesity (DIO) mouse model. Adult male mice were first fed a high-fat diet for six weeks, followed by a standard diet with or without BHB supplementation for an additional six weeks. BHB supplementation led to significant body weight loss independent of food intake. This weight reduction was associated with decreased adipocyte differentiation, reflected by reduced peroxisome proliferator-activated receptor gamma (PPARγ) protein levels and lower uncoupling protein 1 (UCP1) expression, indicating altered SAT function. Transcriptomic analysis of SAT revealed upregulation of genes involved in fatty acid activation and transport (e.g., *Slc27a2*, *Plin5*, *Acot4*, *Acsm3*, *Rik*). Functional enrichment highlighted the activation of the PPAR signaling pathway and enrichment of peroxisomal components in the BHB group. Together, these results suggest that BHB promotes lipid remodeling in SAT, enhancing fatty acid metabolism while suppressing thermogenic pathways, and thus may represent a novel mechanism contributing to adiposity reduction and metabolic improvement.

## 1. Introduction

Obesity has emerged as one of the most significant global health challenges of the 21st century [1]. Characterized by excessive fat accumulation, it represents a major risk for numerous metabolic disorders, including type 2 diabetes, cardiovascular diseases and non-alcoholic fatty liver disease [2]. One of the tissues most affected by obesity is the adipose tissue, particularly white adipose tissue (WAT) and brown adipose tissue (BAT). As obesity develops, WAT undergoes hypertrophy (increased adipocyte size) and hyperplasia (increased adipocyte number) [3]. These changes contribute to insulin resistance, chronic low-grade inflammation, and impaired lipid metabolism, the key drivers of metabolic disorders [4].

In the early stages of obesity, fat primarily accumulates in subcutaneous adipose tissue (SAT) [5]. However, as the condition progresses, visceral fat increases, significantly exacerbating the risk of metabolic complications [6]. In parallel, BAT, a key player in non-shivering thermogenesis [7], becomes functionally impaired, leading to a reduction in energy expenditure and further promoting metabolic dysfunction [8]. BAT is rich in mitochondria and uniquely expresses UCP1, allowing it to dissipate energy as heat, thereby contributing to overall energy homeostasis and metabolic health [9].

One of the most widely explored strategies to counteract obesity-associated adipose tissue dysfunction involves weight loss interventions, which not only reduce fat mass but can also induce the browning (or beiging) of WAT [10]. This process, which primarily occurs in SAT, involves the acquisition of BAT-like features by white adipocytes [10], including the upregulation of UCP1 and other thermogenic genes. The induction of beiging has been identified as a promising therapeutic target for enhancing energy expenditure and improving metabolic health. Cold exposure [11], physical exercise [12], and specific metabolic signals, such as AMP-activated protein kinase (AMPK) activation [13], have been shown to stimulate browning, thereby promoting thermogenesis and mitochondrial activity [14].

At the molecular level, beiging is characterized by the activation of key transcription factors, such as PPARɣ and PR domain-containing 16 (PRDM16), which drive beige adipocyte differentiation, as well as UCP1, which facilitates the dissipation of energy as heat. This metabolic reprogramming of SAT towards a more thermogenic phenotype represents an important mechanism for counteracting obesity-related metabolic impairments, as it enhances caloric expenditure and insulin sensitivity while reducing lipid accumulation in metabolically harmful depots [15,16,17]. Emerging dietary approaches such as the ketogenic diet (KD) and time-restricted eating have gained popularity in promoting significant weight loss by reducing calorie intake and increasing energy expenditure [18,19]. Both strategies share a common feature: the induction of ketonemia, characterized by elevated blood concentration of ketone bodies [18]. Ketone bodies are known to enhance mitochondrial activity by facilitating long-chain fatty acids’ breakdown through β-oxidation [20], which boosts energy production [21,22]. Among these ketone bodies, BHB has been shown to stimulate mitochondrial biogenesis and improve oxidative phosphorylation, making it essential for maintaining mitochondrial health and metabolic flexibility [21]. In addition, BHB activates signalling pathways that support lipid metabolism and energy production during periods of nutrient scarcity [23]. Despite these advances, significant gaps remain in understanding the specific roles and mechanisms of BHB in metabolism.

While BAT activation by BHB has been previously reported [24,25,26], its potential involvement in SAT beiging has not yet been fully investigated. Here, we identified the putative induction of fatty acid oxidation in SAT after BHB supplementation and the regulation of the PPARɣ signalling pathway, highlighting its promise as a clinical strategy for treating obesity and its associated metabolic disorders.

## 2. Results

### 2.1. BHB Supplementation Reduces Weight Gain and Alters Serum Metabolite Profiles Independent of Food Intake in DIO Mice

After 6 weeks on a high-fat diet, mice showed a significant increase in body weight from an initial mean weight of 22 g to 36.38 g (65.36%) (Figure 1A, DIO; *p* < 0.0001). Over the following six-weeks, BHB-treated mice showed a significant increase in weight loss compared to controls. On average, BHB-treated mice lost 7.03 g (19.5%), whereas control mice lost 5.56 g (15.1%), indicating that BHB supplementation led to an additional 1.47 g of weight loss. This corresponds to a 4.4% greater relative reduction in body weight and, in absolute terms, a 26.4% greater weight loss compared to the control group (Figure 1A, BHB; *p* < 0.01), despite no significant differences in accumulated food intake (controls: 568 ± 7.033 g; BHB: 554.4 ± 2.283 g). Serum metabolites were analysed by ^1^H NMR spectroscopy, revealing notable decreases in branched-chain amino acids (BCAAs)—specifically lysine, isoleucine, and valine (Figure 1B–D, *p* < 0.05)—along with a significant increase in key energy metabolism intermediates such as pyruvate (Figure 1E, *p* < 0.001), citric acid (Figure 1F, *p* < 0.01) and BHB; controls: 48.8 ± 1.86 µM; BHB: 55.37 ± 2.19 µM; *p* < 0.05).

### 2.2. BHB Supplementation Regulates the UCP-1-Dependent Thermogenic Pathway in the SAT of DIO Mice

*Ucp1* mRNA expression was evaluated in BAT to determine whether or not BHB could be involved in a classical thermogenic pathway. BHB-treated mice exhibited a significant reduction in *Ucp1* mRNA expression in BAT compared to controls (Figure 2A; *p* < 0.05). To explore the implication of BHB supplementation on WAT beiging, *Ucp1* mRNA expression was also analysed in visceral adipose tissue (VAT) and SAT. No changes were found between groups (Figure 2A).

Focusing on SAT, different thermogenic markers (*Pgc-1α* and *Pparγ*) were analysed. *Pgc-1α* mRNA expression was significantly elevated in the BHB group compared to the control group (Figure 2B; *p* < 0.01), while no significant changes were found in the mRNA levels of *Pparɣ* (Figure 2B).

To corroborate these results, we analysed the protein levels of these markers in SAT. Interestingly, PGC-1α protein levels remained unchanged (Figure 2C,D), but PPARɣ and UCP1 protein levels were reduced in BHB compared to controls (Figure 2C,D; *p* < 0.01). These results suggest a complex regulatory mechanism for thermogenesis in SAT.

### 2.3. BHB Supplementation Induced Differential Expression of SAT Fatty Acid Oxidation Genes in DIO Mice

To unveil the mechanism underlying the benefits of BHB supplementation in maintaining WAT energy homeostasis during obesity, we performed RNAseq in SAT from DIO mice. This allowed us to analyse differentially expressed genes involved in thermogenesis and energy expenditure pathways.

Differentially expressed genes (DEGs) between groups are shown in the volcano plot (Figure 3A) and total DEG counts (Figure 3B). A total of 15,000 genes passed the expression filter, but only 8 genes were prevalent DEG (*p* < 0.05), including 6 upregulated (*Acot4*; *Acsm3*; *Slc272a*; *Plin5*; *Fam151a*; 2310069b03 *RIK*) and 2 downregulated genes (*Egf*; *Igkv1-99*).

To gain insight into the functional relevance of the DEGs in subcutaneous adipose tissue following BHB supplementation, we conducted enrichment analyses using Reactome, Gene Ontology (GO) in all three aspects (Biological Process (BP), Cellular Component (CC), Molecular Function (MF)) and Kyoto Encyclopedia of Genes and Genomes (KEGG) databases.

BP (Figure 3C) highlighted upregulation in genes associated with peroxisomal activity, β-oxidation, and lipid metabolism. CC (Figure 3D), revealed significant associations with key organelles involved in lipid and energy metabolism, including peroxisomes, mitochondria, endoplasmic reticulum, and lipid droplets, as well as components related to secretion and signaling, such as extracellular exosomes. MF (Figure 3E) further revealed enrichment in fatty acid-CoA ligase activities (e.g., *Slc27a2*, *Acsm3*), lipid transporters, and hydrolases, indicating increased lipid activation and remodeling. Additionally, pathways related to kinase activation and transmembrane receptor signaling were upregulated, pointing to broader metabolic reprogramming in response to BHB. Accordingly, the PPAR signaling pathway emerged as the most significantly enriched, suggesting a shift toward enhanced fatty acid oxidation (Figure 3F).

### 2.4. BHB Fails to Modulate FAO Through Carnitine Palmitoyltransferase 2 (CPT2) in SAT

Based on the FAO genes differentially expressed in BHB group, we aimed to determine whether BHB supplementation could enhance fatty acid oxidation in SAT, thus potentially improving metabolic health in obesity. Due to its crucial role in the final step of mitochondrial fatty acid transport, ensuring the conversion of acyl-carnitines back into acyl-CoA for β-oxidation, we focused on Carnitine Palmitoyltransferase 2 (CPT2). This key enzyme is involved in the transport of long-chain fatty acids into the mitochondria for subsequent FAO. However, although a slight increase without statistical significance was observed in CPT2 protein levels, these remained unchanged after BHB supplementation (Figure 4A,B).

## 3. Discussion

Recent studies have highlighted ketone bodies as promising modulators of body weight, although the precise molecular mechanisms remain incompletely understood [27]. In the present work, we explored the effects of BHB supplementation on adipose tissue function in a DIO mouse model.

BHB-supplemented mice exhibited significantly greater weight loss over a six-week period, without changes in food intake, suggesting a potential increase in energy expenditure. Given the key role of SAT in thermogenesis via beiging [28], we investigated mechanisms that might underlie these observations.

Serum metabolomic analysis revealed elevated levels of pyruvate and citric acid in the BHB group, suggesting a metabolic state favouring oxidative energy metabolism. As both metabolites are central to mitochondrial function—with pyruvate fueling the TCA cycle and citric acid being a key intermediate—these changes may reflect enhanced mitochondrial activity. However, since ATP production was not directly measured, this remains speculative. Future studies should include direct assessments of mitochondrial respiration and ATP levels to validate these interpretations.

Additionally, BHB supplementation led to a significant reduction in BCAAs, specifically isoleucine and valine. This is consistent with previous research linking lower BCAA levels to improved lipid metabolism in the context of obesity [29,30]. Elevated BCAAs have been associated with insulin resistance and dyslipidemia [31], and their reduction may reflect a beneficial metabolic shift. One possible mechanism underlying the observed metabolic improvements could involve a reduced mobilization or turnover of BCAAs from skeletal muscle, potentially contributing to enhanced systemic metabolic efficiency and insulin sensitivity [30]. However, our study did not assess glucose, insulin, or lipid profiles, which limits mechanistic conclusions. Future work should adopt a broader metabolic profiling approach.

Interestingly, despite reports that ketone bodies stimulate thermogenesis in SAT [32], we found reduced protein levels of Pparγ and Ucp1 following BHB treatment, while mRNA levels remained unchanged. This discrepancy may reflect the epigenetic actions of BHB, which is known to modulate histone acetylation and affect transcription factor activity [33,34]. These effects could alter translation efficiency or protein stability without changing gene expression. Further analysis of mRNA stability and protein turnover is warranted to clarify this.

Our findings contrast with those of Walton et al. [32], who reported increased Ucp1 and Pgc-1α mRNA levels following a ketogenic diet, though without significant body weight changes. Differences in duration, delivery route (diet vs. drinking water), and model (DIO vs. chow-fed mice) likely explain the discrepancies. Moreover, our data suggest that short-term BHB supplementation primarily affects SAT, with minimal changes in BAT or VAT, consistent with the notion that SAT is the first depot to respond to obesity-induced changes [5] and considering that a 6-week supplementation might be a relatively short-term treatment compared to previous studies reporting BAT activation [24,25].

Transcriptomic analysis of SAT revealed upregulation of genes involved in mitochondrial fatty acid β-oxidation, including *Plin5*, *Slc27a2*, *Acsm3*, and *Acot4*. These genes facilitate triglyceride mobilization and oxidation, helping to prevent mitochondrial lipid overload [35,36,37,38]. Interestingly, BHB also increased expression of the long non-coding RNA *2310069B03Rik*, a negative regulator of UCP1 [39], suggesting complex regulation of thermogenesis. Downregulation of *Egf*, a gene involved in lipid storage and adipocyte differentiation [40], further supports a metabolic shift toward lipid catabolism.

Notably, *Cpt2* expression, which governs the final step of mitochondrial fatty acid import [41,42,43], was unchanged. In line with these findings, gene fuctional enrichment analyses using Reactome, GO and KEGG databases revealed that BHB could modulate upstream lipid metabolism, favouring alternative pathways such as peroxisomal β-oxidation, as suggested by increased *Acot4* expression [44], enrichment of peroxisomal processes, and PPAR signaling.

Altogether, these results underscore the nuanced balance between mitochondrial and peroxisomal FAO in SAT, and the potential metabolic flexibility induced by BHB.

## 4. Materials and Methods

### 4.1. Animals

Eight-week-old C57BL/6 male mice (n = 20) were used in these experiments. The animals were housed in the animal facility of the Faculty of Medicine in Malaga under controlled conditions of light (14 h, from 7:00 a.m.) and temperature (22 °C), with free access to tap water and food. The animals were inspected daily during experimental interventions by qualified personnel to monitor for any health issues. All experiments and animal protocols were reviewed and approved by the local Ethics Committee and complied with Royal Decree 1201/2005 (BOE n° 252) regarding the protection of experimental animals and with the Directive of the Council of the European Communities (86/609/EEC). The experiments were conducted in accordance with the guidelines and protocols of the Royal Decree 53/2012 on the care and protection of animals used for scientific purposes. Body weight and food intake were monitored weekly throughout the experiment. No adverse events were observed.

### 4.2. Experimental Design: Diet-Induced Obesity (DIO) Mouse Model and BHB Supplementation

Mice were fed with a high-fat diet over 6 weeks to induce an obesity phenotype. The composition of the high-fat diet (HFD) used in this study was as follows (per 773.85 g of diet): Protein (Casein, Lactic, 30 Mesh: 200.00 g, L-Cystine: 3.00 g); Carbohydrate (Lodex 10: 125.00 g, Sucrose, Fine Granulated: 72.80 g); Fiber (Solka Floc (FCC200): 50.00 g). Fat (Lard: 245.00 g, Soybean Oil (USP grade): 25.00 g); Minerals (Mineral mix (S10026B): 50.00 g); Vitamins (Choline bitartrate: 2.00 g, Vitamin mix (V10001C): 1.00 g); Dye Blue FD&C #1 Alum; Lake (35–42%): 0.05 g; (HFD; Diet D12492, 60% fat content; Brogaarden ApS). Subsequently, a chow diet was introduced to study the isolated effect of BHB supplementation. The mice were randomly divided into two equal groups. One group received BHB supplementation (KetoForce food supplement) at a concentration of 0.08 g/mL, corresponding to a final concentration of 4.2% BHB salt, administered in the drinking water for 6 weeks (BHB group; n = 10). The control group (control; n = 10) received plain drinking water. To prevent compound precipitation, fresh BHB solution was prepared daily.

### 4.3. Tissue Collection

At the end of the experiment, animals were euthanized via cervical dislocation in accordance with laws and regulations for animal experimentation. The extracted tissues included BAT, SAT and VAT. All samples were stored at −80 °C until further processing.

### 4.4. RT-qPCR

Total RNA was extracted from SAT, VAT, and BAT using a Nucleospin^®^ RNAII kit (Macherey–Nagel GmbH, Düren, Germany). RNA concentration and purity were assessed using a Nanodrop 2000c spectrophotometer (Thermo Scientific, Waltham, MA, USA). A total of 1 µg of total RNA was reverse-transcribed into cDNA using a high-capacity cDNA archive kit (Applied Biosystems, Loughborough, UK) following the manufacturer’s instructions. Real-time PCR was performed using Kapa probe fast (VWR, KK4702) on duplicate samples with a predesigned Taqman gene expression assay for *Pgc-1α* (Mm01208835_m1), *Pparγ* (Mm00440940_m1) and *Ucp1* (Mm01244861_m1) using an CFX 96 Real-Time System. Samples were assayed with *S18* (VIC™/MGB probe, Hs99999901_s1, ThermoFisher) as the endogenous control for normalization. Gene expression was calculated relative to the endogenous control samples and to the control sample, giving an RQ value (2^−ΔΔCt^ method, where Ct represents the cycle threshold).

### 4.5. RNA Sequencing Analysis (RNAseq)

Library Construction, Quality Control and Sequencing (Novogene, Beijing, China)

mRNA was purified from total RNA using poly-T oligo-attached magnetic beads. After fragmentation, the first-strand cDNA was synthesized using random hexamer primers, followed by second-strand cDNA synthesis using either dUTP for the directional library or dTTP for the non-directional library [45]. For the non-directional library, workflow included end repair, A-tailing, adapter ligation, size selection, amplification, and purification. The directional library required end repair, A-tailing, adapter ligation, size selection, USER enzyme digestion, amplification, and purification. The resulting libraries were validated using Qubit and real-time PCR for quantification and a bioanalyzer for size distribution detection. Quantified libraries were pooled and sequenced on Illumina platforms (PE150; CASAVA v1.8.2 analysis software) according to effective library concentration and data amount, yielding fastq files filtered for low-quality, N-rich, or adaptor-polluted reads. Reads from the RNA-seq libraries were filtered to remove adapters or low-quality reads. The resulting high-quality reads were used for downstream RNA-seq analysis.

#### Bioinformatics Analysis Pipeline

For the differential expression analysis of RNA-seq, the raw sequences from 28 samples were first pre-processed and mapped using the pipeline described by Córdoba-Caballero et al. [46]. Preprocessing to remove low-quality reads and technical artifacts was performed with SeqtrimBB (based on the BBmap suite), configured to a minimum read length of 135 nucleotides and a minimum sequence quality (min qual) of 20. The pre-processed reads were aligned to the mouse genome assembly mmGRcm38. Alignment and count table generation for each sample were performed using STAR (version 2.5.3a) [47]. Differential expression analysis was performed using ExpHunter Suite [48]. We set ExpHunter Suite to use the DESeq2 algorithm with a log2FC of 1 and a *p*-value of 0.05, filtering out lowly expressed genes to a minimum of two counts per million mapped reads in at least two samples per group. Differentially expressed genes resulting for each comparison were used for functional analysis, selecting the Gene Ontology, KEGG and Reactome databases, with a minimum *p*-value set to 0.1.

### 4.6. Western Blot

Protein from SAT was extracted from the organic phase using the Nucleospin^®^ RNAII kit (Macherey-Nagel, GmbH, Düren, Germany) and precipitated by adding 4 volumes of ice-cold acetone followed by incubation over night at −20 °C. Finally, proteins were centrifuged for 10 min at max speed, and the pellet was resuspended in lysis buffer (Urea 70 mM, Thiourea 2 M, 4% 3-[(3-Cholamidopropyl); dimethylammonio]-1-propanesulfonate (CHAPS), Sigma; and Tris 1M (pH 8.8). Total protein quantification was carried out using the Bradford Assay (BioRad, Hercules, CA, USA). An equal amount of protein was mixed with 4× SDS sample buffer (Tris-HCl 100 mM, pH 6.8, 4% SDS, 2% bromophenol blue, 20% glycerol; Sigma, Mildenhall, UK) and boiled at 96 °C for 5 min. After resolving on SDS-polyacrylamide gel electrophoresis (SDS-PAGE), gels were transferred to PVDF membranes (Millipore, Burlington, MA, USA) previously activated with 100% methanol for 5 min. Then, membranes were blocked using 5% non-fat dried milk (Santa Cruz Biotech, Dallas, TX, USA) in TTBS (10 mM Tris-HCl, 150 mM NaCl pH 7.5, Tween 20 (0.05% *w*/*v*)) for 2 h and incubated overnight at 4 °C with primary antibody: PPARɣ (1:1000; Cat#MA5-14889); PGC-1α 1:1000; Cat#ab54418); UCP1 (1:1000; Cat#ab10981); CPT2 (1:1000; Cat#265551AP) or mouse anti-actin (1:5000; Cat#A5316, Sigma, UK).The membranes were then washed and incubated (RT, 2 h) with a secondary horseradish peroxidase-linked anti-rabbit (1:2000 in blocking buffer; 111-035-003, Jackson, Glendora, CA, USA) or anti-mouse (1:10,000 in blocking buffer; 115-035-003, Jackson, USA) antibody. Immunoreactive bands were visualized in a ChemiDoc™ MP digital imager (Bio-Rad Laboratories, Hercules, CA, USA) by using WesternBright enhanced chemiluminescent substrate (BioRad, Hercules, CA, USA). Experiments were quantified by densitometry using ImageJ software (version 2.14.0, National Institutes of Health, Bethesda, MD, USA, https://imagej.net/ij/ accessed on 20 November 2023).

### 4.7. Metabolite Quantification by ^1^H NMR Spectroscopy

All spectra were acquired using a Bruker AVANCE^TM^ 600 MHz spectrometer (Bruker BioSpin, Ettlingen, Germany) equipped with an *Advance III* console and either a 4 mm TXI HR-MAS probe for intact tissue analysis or a nitrogen-cooled TCI Prodigy cryoprobe for plasma sample analysis.

For intact tissue samples, water-suppressed ^1^H high-resolution magic angle spinning (HR-MAS) NMR spectra were acquired using a Carr–Purcell–Meiboom–Gill (*CPMG*) sequence with the following parameters: 12 kHz spectral width, 64 k data points, 64 scans, 1 ms echo time (2τ) with a total echo time of 130, and 5 s relaxation delay. Water pre-saturation was applied during the relaxation delay. Metabolite quantification was performed with the software LCModel (Version 6.3-0E) [49], as described elsewhere [50], using creatine as the internal reference.

For plasma samples, acquisition sequence included 1D Nuclear Overhauser Effect SpectroscopY (NOESY), 1D Carr–Purcell–Meiboom–Gill (CPMG) for T_2_ editing of macromolecule signals, and 2D J-resolved (JRES) to aid in the identification of metabolites. Small metabolites were quantified on the CPMG spectra using the software Chenomx (v10.0, Chenomx Inc., Edmonton, AB, Canada), with the electronic reference to access in vivo concentration (ERETIC), as implemented in TopSpin 3.5.pl7 (ERETIC 2), serving as the concentration reference. 

### 4.8. Statistical Analysis

Statistical analyses were conducted using Prism software (GraphPad Prism version 10.1.2 for Windows, GraphPad Software, San Diego, CA, USA, EE.UU.) Data are expressed as mean ± SEM. Normality was initially assessed, and homogeneity of variance was evaluated using the F test. No additional methods were used to further validate statistical assumptions. For group comparisons, Student’s *t*-test (unpaired, two-tailed) was performed, and *p*-values < 0.05 were considered statistically significant.

## 5. Conclusions

Collectively, our findings highlight a novel regulatory role of BHB in modulating lipid mobilization and β-oxidation in SAT, providing insights into how these metabolites influence energy balance beyond conventional thermogenesis pathways. These data suggest that ketone bodies may optimize lipid metabolism in SAT by enhancing fatty acid turnover and mitochondrial function, potentially contributing to the observed reductions in weight gain.

## 6. Limitations of the Study

This study provides novel insights into SAT metabolism in response to BHB supplementation; however, several limitations must be acknowledged. First, gene expression analysis was limited to SAT. Including BAT and VAT would offer a more comprehensive understanding of depot-specific metabolic adaptations. Moreover, while RNA sequencing revealed relevant transcriptional changes, further validation by qPCR and protein-level analyses is needed but was not feasible due to sample constraints.

All experiments were performed in male mice to avoid hormonal variability. Future studies should include females to explore potential sex-specific responses, as sex can influence ketone metabolism and lipid oxidation. Additionally, the six-week duration limits conclusions about long-term effects, and the absence of direct thermogenic assessments (e.g., calorimetry, thermal imaging) constrains interpretation of energy expenditure.

Lastly, although our findings suggest promising metabolic effects of BHB, translational relevance must be approached cautiously. Rodent and human metabolic responses differ, and clinical trials are needed to define optimal dosing, duration, and safety of BHB supplementation in humans. Exploring its combination with exercise or dietary strategies may further enhance its therapeutic value.

## Figures and Tables

**Figure 1 ijms-26-05064-f001:**
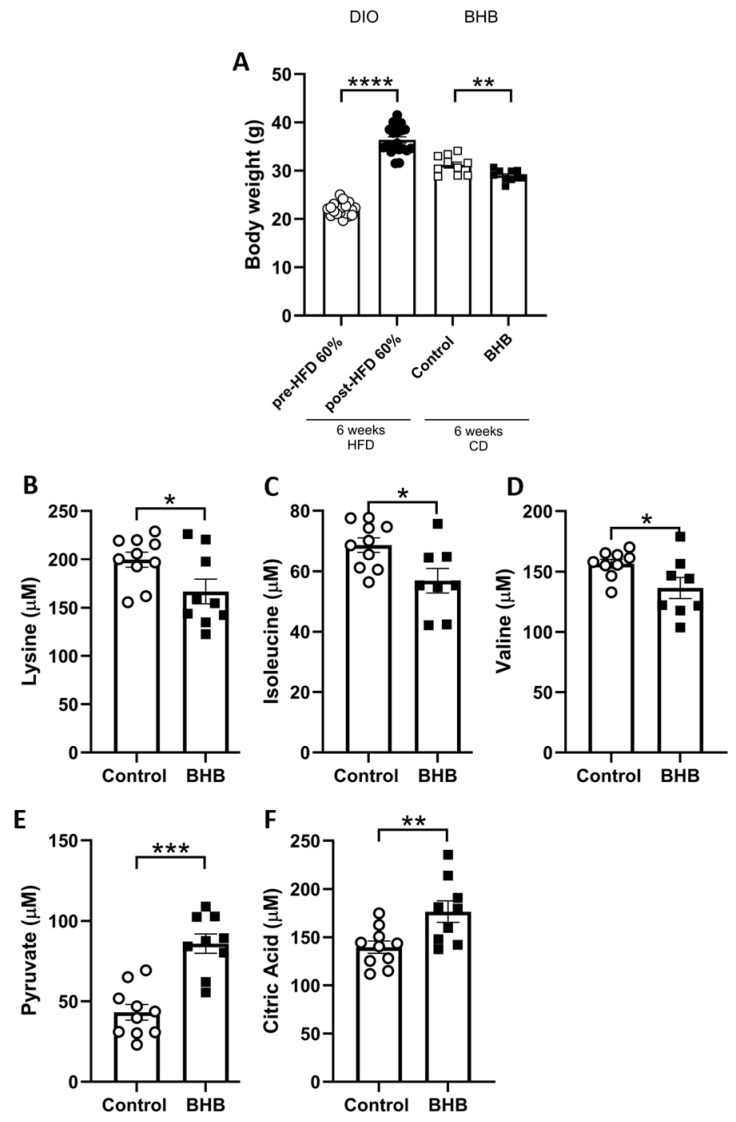
BHB supplementation reduces body weight gain without affecting food intake and modulates serum metabolite levels in a DIO mouse model. Male C57BL/6J mice were fed a high-fat diet (HFD; 60% fat) for 6 weeks to induce obesity, creating a diet-induced obesity (DIO) model. Following this induction period, the mice received water supplemented with BHB) for an additional 6-weeks to assess the effects of ketone body supplementation. (**A**) Body weight (g) pre- and post-60% high-fat diet and body weight (g) pre- and post-treatment, showing a significant reduction in the BHB-supplemented group. To investigate the metabolic effects of BHB supplementation in obesity, we analysed the serum concentrations of key metabolites associated with BCAAs amino acids and energy metabolism. Serum concentration of (**B**) lysine, (**C**) isoleucine, (**D**) valine were significantly decreased, whereas (**E**) pyruvate and (**F**) citric acid concentrations were elevated in DIO mice following BHB supplementation. Data are expressed as the mean (±SEM). Student’s *t*-test was performed (*n* = 8–10). * *p* < 0.05, ** *p* < 0.01, *** *p* < 0.001, **** *p* < 0.0001.

**Figure 2 ijms-26-05064-f002:**
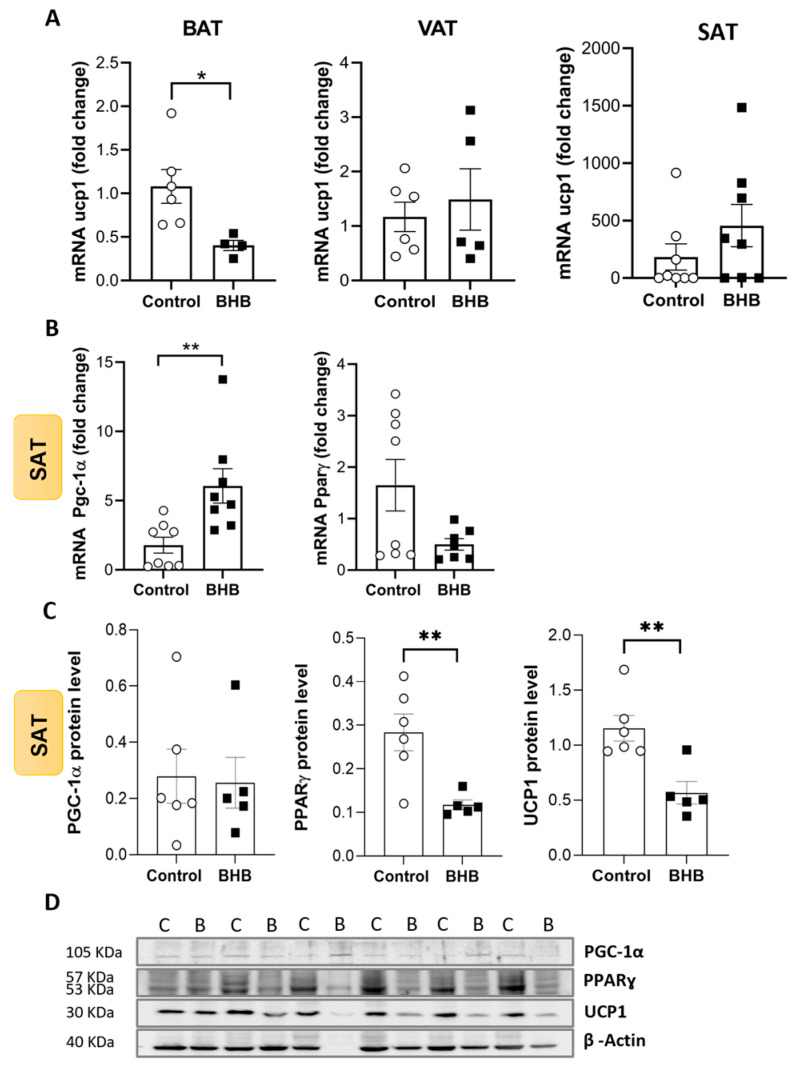
BHB modifies thermogenic markers in adipose tissue of DIO mice. The mRNA expression levels of key thermogenic genes were analysed in BAT, VAT, and SAT to better understand the effect of BHB on the regulation of thermogenic pathways during obesity. (**A**) mRNA expression level of *Ucp1* in BAT was significantly reduced in BHB mice vs. controls, while mRNA expression levels of *Ucp1* in VAT and SAT remained unchanged. (**B**) mRNA expression level of *Pgc-1α* in SAT was significantly higher in BHB mice vs. controls, while mRNA expression levels of *Pparɣ* in SAT of BHB group were not modified. (**C**) Protein levels of PGC-1α were not altered, while PPARɣ and UCP1 protein levels were diminished in SAT after BHB supplementation. (**D**) Immunoblot images are shown. Data are expressed as mean (±SEM). Student’s *t*-test was performed. (*n* = 4–8). (C: control; B: β-Hydroxybutyrate). * *p* < 0.05, ** *p* < 0.01.

**Figure 3 ijms-26-05064-f003:**
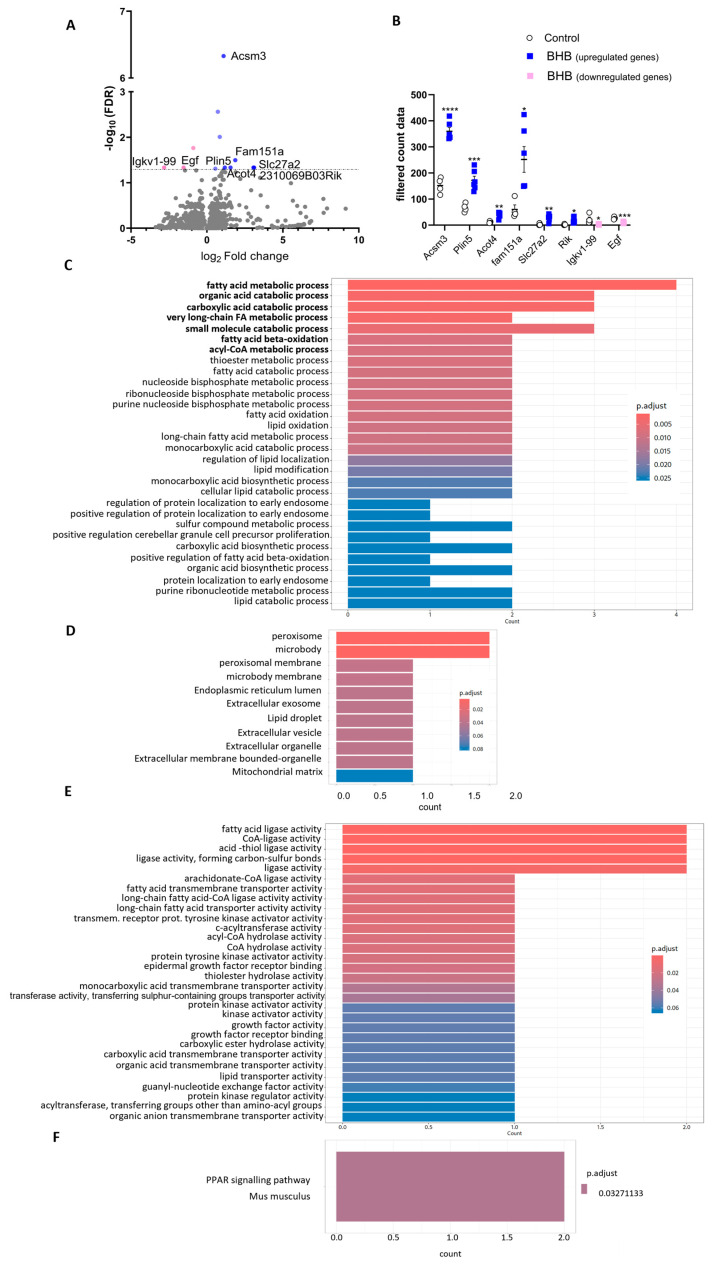
Identification of differentially expressed genes related to fatty acid oxidation (FAO), thermogenesis in SAT of DIO mice supplemented with BHB compared to control group. RNA sequencing (RNA-seq) analysis was performed on SAT comparing the BHB-supplemented group with the control group to identify changes in gene expression (**A**) Volcano plot displaying differentially expressed genes between the two experimental groups. Pink dots represent significantly downregulated genes, while blue dots represent significantly upregulated genes in SAT. Genes in grey have no statistical significance. The horizontal dashed black line indicates an adjusted *p*-value threshold of 0.05. (**B**) Filtered count data for selected upregulated (blue squares) and downregulated (pink squares) genes compared to control group (white circles), derived from the volcano plot. The *y*-axis represents normalized expression counts, and the *x*-axis lists individual genes grouped by their regulation status (upregulated or downregulated). (**C**) Gene Ontology over-representation analysis in regards to Biological process (**D**), Cellular components and (**E**) Molecular function showing the most significantly enriched pathways ranked by adjusted *p*-value in ascending order. The *x*-axis represents the number of significant genes identified within each functional category. (**F)** KEGG pathway enrichment analysis highlighting the activation of the PPAR signalling pathway based on differentially expressed genes, performed using ExpHunter Suite (v. 1.16.0) (*n* = 4–8). Data are expressed as the mean (±SEM). Student’s *t*-test was performed. * *p* < 0.05, ** *p* < 0.01, *** *p* < 0.001, **** *p* < 0.0001.

**Figure 4 ijms-26-05064-f004:**
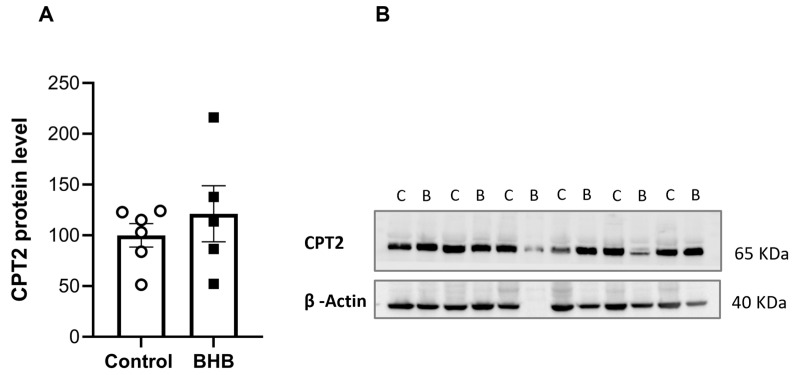
CPT2 levels were not significantly altered in SAT of DIO mice supplemented with BHB. (**A**) CPT2 protein expression was analysed in SAT from C57BL/6 DIO mice supplemented with BHB and control groups, with (**B**) Representative immunoblot images of CPT2 protein levels are shown. BHB supplementation did not significantly change the levels of CPT2 compared to controls. Data are presented as the mean (±SEM). Student’s *t*-test was performed. (*n* = 6). (C: control; B: β-Hydroxybutyrate).

## Data Availability

Datasets used in this study are available from the corresponding author upon request.

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
