# Peer review of "β-Hydroxybutyrate Reduces Body Weight by Modulating Fatty Acid Oxidation and Beiging in the Subcutaneous Adipose Tissue of DIO Mice"

_ijms, 2025, doi:10.3390/ijms26115064_

Round 1

Reviewer 1 Report

Comments and Suggestions for Authors

In this manuscript, Violeta Heras and colleagues report that BHB modulates lipid metabolism in subcutaneous adipose tissue (SAT), potentially contributing to reduced adiposity through mechanisms distinct from classical thermogenesis. Overall, the study is interesting and relevant; however, several issues should be addressed to improve the clarity and rigor of the work:

Abstract: The abstract predominantly describes background information and methodology, but lacks a clear presentation of the key results. Please incorporate major findings into the abstract.

Introduction: The acronym "SAT" is explained twice—first as "subcutaneous adipose tissue" (line 60) and later as "subcutaneous WAT" (line 69). Please review and ensure consistent terminology throughout.

Figure 2D: Based on the actin expression levels, it is difficult to assert that there are meaningful differences in UCP1 between groups, particularly for wells 2, 8, and 10. Additionally, the actin band in well 6 appears inconsistent.

Figure 3: Typically, GO analysis includes results for Cellular Component (CC), Biological Process (BP), and Molecular Function (MF). Why is only BP presented here? Furthermore, for KEGG analysis, why is only a single pathway shown? Please clarify or expand.

Figure 4: Similar to Figure 2D, the Western blot results show variability across the BHB-treated groups, with some samples showing increased and others decreased expression. Thus, it is difficult to conclude that BHB consistently promotes CPT2 expression.

Reviewer 2 Report

Comments and Suggestions for Authors

Ketone bodies are alternative metabolites for energy production. However, their contribution to metabolic reprogramming is not well understood. Heras et al., 2025 provide data suggesting that treatment with the ketone body beta-hydroxybutyrate (BHB) can reduce body weight by modulating fatty acid oxidation and beiging in subcutaneous adipose tissue of DIO mice. Interestingly, the findings suggest that BHB modulates lipid metabolism in subcutaneous adipose tissue contributing to reduced adiposity through mechanisms distinct from classical thermogenesis.

Research on the role of ketone bodies in energy metabolism is up-to-date. The manuscript is well written and the findings are novel and interesting. Minor revisions could be done.

Specific points

  1. The dashed line in figure 1A could be removed. Figure 1E, Y axis, piruvate should be corrected.
  2. The exact composition of the high-fat diet should be indicated in materials & methods section.

Round 2

Reviewer 1 Report

Comments and Suggestions for Authors

The authors have addressed all previous concerns, and the manuscript is now suitable for publication.